# β-Peltoboykinolic Acid from *Astilbe rubra* Attenuates TGF-β1-Induced Epithelial-to-Mesenchymal Transitions in Lung Alveolar Epithelial Cells

**DOI:** 10.3390/molecules24142573

**Published:** 2019-07-15

**Authors:** In Jae Bang, Ha Ryong Kim, Yukyoung Jeon, Mi Ho Jeong, Yong Joo Park, Jong Hwan Kwak, Kyu Hyuck Chung

**Affiliations:** 1School of Pharmacy, Sungkyunkwan University, Suwon 16419, Korea; 2College of Pharmacy, Daegu Catholic University, Gyeongsan 38430, Korea

**Keywords:** *Astilbe rubra*, β-peltoboykinolic acid, epithelial-mesenchymal transition, TGF-β1, lung fibrosis

## Abstract

Epithelial-to-mesenchymal transition (EMT) is increasingly recognized as contributing to the pathogenesis of idiopathic pulmonary fibrosis. Therefore, novel plant-based natural, active compounds have been sought for the treatment of fibrotic EMT. The aim of the present study was to investigate the inhibitory effects of *Astilbe rubra* on TGF-β1-induced EMT in lung alveolar epithelial cells (A549). *A. rubra* was subjected to extraction using 70% ethanol (ARE), and ethanol extracts of the aerial part and that of the rhizome were further partitioned using various solvents. Protein expression and cell motility were investigated to evaluate the inhibitory effects of ARE on EMT. EMT occurred in A549 cells treated with TGF-β1, but was prevented by co-treatment with ARE. The dichloromethane fractions showed the strongest inhibitory effect on TGF-β1-induced EMT. β-Peltoboykinolic acid was isolated from the dichloromethane fractions of *A. rubra* by activity-oriented isolation. β-Peltoboykinolic acid not only attenuated TGF-β1-induced EMT, but also the overproduction of extracellular matrix components including type I collagen and fibronectin. The Smad pathway activated by TGF-β1 was inhibited by co-treatment with β-peltoboykinolic acid. Taken together, these results indicate that β-peltoboykinolic acid from *A. rubra* and dichloromethane fractions shows potential as an antifibrotic agent in A549 cells treated with TGF-β1.

## 1. Introduction

Lung fibrosis is a chronic disease characterized by the accumulation of extracellular matrix (ECM) and a progressive decline in lung function. Persons diagnosed with idiopathic pulmonary fibrosis (IPF) have a poor prognosis, with a median survival of less than three years [1]. Prior efforts to treat IPF have included anti-inflammatory therapy, which has not proven to be effective. Recent findings indicate the pathogenesis of lung fibrosis may be epithelial-driven [2]. There are numerous studies showing that abnormally activated lung epithelial cells contain signaling proteins responsible for the migration, proliferation, and activation of fibroblasts. In addition, epithelial cells contribute to the expansion of the fibroblast/myofibroblast population through epithelial-to-mesenchymal transition (EMT) [3]. Epithelial plasticity with partial EMT reprogramming might be central to wound repair, but under chronic and unresolved stress, this response can be disrupted, leading to fibrosis [4]. Therefore, EMT regulation is a novel strategy for the suppression of fibrosis, and chemotherapeutics able to target this event show great promise.

EMT is a dynamic cellular process where polarized epithelial cells lose their epithelial phenotype and gain mesenchymal characteristics. Dissociation of cell–cell junctions and acquisition of mesenchymal markers are important signs of EMT [5]. Transforming growth factor-β1 (TGF-β1) is thought to act as a master switch in EMT. TGF-β1 primarily triggers EMT via a canonical Smad-dependent pathway, which requires two types of receptor kinases and a family of signal transducers called Smad2/3. Upon phosphorylation, Smad2/3 forms complexes with Smad4, and subsequently translocates to the nucleus to regulate the transcription of downstream genes responsible for EMT [6]. Snail, which comprises downstream transcription factors in the Smad pathway, directly combines with the promoter of the E-cadherin encoding gene, *CDH1*, to silence E-cadherin expression [7].

*Astilbe rubra* Hook. f. et Thomas. (family Saxifragaceae) is a perennial herbaceous plant that grows in mountainous areas of Korea. The whole plant has been used to treat contusion, arthralgia, and stomachalgia in traditional Chinese medicine, and is also used as a food source [8]. To date, it has been reported to exhibit antioxidant effects, inhibitory activity against α-amylase, α-glucosidase, and lipase enzymes, and anti-cancer effects [9,10,11]. However, the effect of *A. rubra* on lung fibrosis has not yet been investigated. The aim of the present study was to evaluate the inhibitory effect of *A. rubra* on EMT. First, we examined whether the 70% ethanol extract derived from the *A. rubra* whole plant inhibited EMT. Second, *A. rubra* was divided into two parts, the aerial part and rhizome, and subjected to extraction with 70% ethanol; the extract was then partitioned with organic solvents. Activity-guided separation of dichloromethane fractions from *A. rubra* extracts led to the isolation of β-peltoboykinolic acid, which effectively inhibited EMT. In addition, the inhibitory effects of β-peltoboykinolic acid on the production of type I collagen (COL1) and fibronectin, which are the major components of the ECM in fibrous tissues, were identified. Finally, the mechanism underlying the inhibition of EMT induction and ECM overproduction was investigated by assessing the effect of β-peltoboykinolic acid on the Smad pathway activated by TGF-β1.

## 2. Results

### 2.1. Ethanol Extract of A. rubra Whole Plant Inhibits TGF-β1-Induced EMT in A549 Cells

A549 cell viability of over 80% was observed after treatment with 125 μg/mL of ARE (Appendix A). A549 cells were co-treated with 2 ng/mL TGF-β1 and various concentrations of ARE (25, 50, 75, 100, and 125 µg/mL). TGF-β1 increased the expression of N-cadherin and vimentin, and decreased the expression of E-cadherin when compared to that in the control. However, these changes in protein expression induced by TGF-β1 were prevented by ARE (Figure 1A). Mobility tended to differ between the control and TGF-β1-treated cells, which indicated that TGF-β1 induced migration in A549 cells. ARE effectively decreased the migration induced by TGF-β1 (Figure 1B). Collectively, these results demonstrate that ARE inhibited TGF-β1-induced EMT in A549 cells.

### 2.2. Dichloromethane Fractions Show the Strongest Inhibitory Effect on TGF-β1-Induced EMT

Each extract from the aerial part and rhizome of *A. rubra* was suspended in distilled water (DW), and the resulting solutions were consecutively partitioned with CH_2_Cl_2_, EtOAc, and n-BuOH to yield four solvent fractions (Figure 2). All solvent fractions from the extracts of the rhizome and aerial parts at a concentration of 100 μg/mL, except for the CH_2_Cl_2_ fractions from the extracts of the aerial part and rhizome of *A. rubra* (ARADF and ARRDF, respectively), resulted in a cell viability of over 80%. For ARADF and ARRDF, a cell viability of over 80% was obtained at a treatment concentration of 50 μg/mL (Appendix A). Both ARAE and ARRE inhibited TGF-β1-induced EMT (Figure 3A). Among the fractions, ARAEF, ARAWF, ARREF, and ARRWF did not affect TGF-β1-induced EMT. However, ARADF and ARRDF showed the strongest inhibitory effect on TGF-β1-induced EMT among all tested fractions (Figure 3B).

### 2.3. β-Peltoboykinolic Acid Attenuates TGF-β1-Induced EMT

ARRDF was fractionated using silica gel chromatography. A selected subfraction was re-chromatographed on reversed-phase C_18_ and silica gel using different solvent combinations to yield β-peltoboykinolic acid. β-Peltoboykinolic acid was obtained as a white solid and identified by comparison of its spectral data (Appendix A) with published values [12]. The structure of β-peltoboykinolic acid is shown in Appendix A. As β-peltoboykinolic acid exhibited significant cytotoxicity at more than 20 μg/mL, 10 μg/mL was determined to be the maximum treatment concentration (Appendix A). A549 cells were treated with TGF-β1 (2 ng/mL) alone or with β-peltoboykinolic acid (1, 5, and 10 μg/mL) for 48 h. Treatment with TGF-β1 induced the mesenchymal phenotype including elongated and spindle-like shapes, which were markedly suppressed by treatment with 5 and 10 μg/mL of β-peltoboykinolic acid (Figure 4A). β-Peltoboykinolic acid co-treatment attenuated, in a dose-dependent manner, the decrease in E-cadherin expression and the increase in N-cadherin and vimentin expression induced by TGF-β1 at the protein and mRNA levels (Figure 4B,C). Mobility tended to differ between the control and TGF-β1-treated cells and was effectively decreased by β-peltoboykinolic acid (Figure 4D,E). Collectively, these results demonstrate that β-peltoboykinolic acid attenuated TGF-β1-induced EMT in A549 cells. We evaluated whether β-peltoboykinolic acid affected ECM overproduction. The protein and mRNA expression of COL1 and fibronectin induced by TGF-β1 was decreased by β-peltoboykinolic acid in a dose-dependent manner (Figure 5).

### 2.4. β-Peltoboykinolic Acid Interrupts the Activation of the Smad Pathway by TGF-β1

The Smad pathway was investigated to elucidate the inhibitory mechanism of β-peltoboykinolic acid on EMT and fibrotic responses induced by TGF-β1. At the protein level, the decrease in Smad expression by TGF-β1 was restored by β-peltoboykinolic acid. TGF-β1-stimulated A549 cells showed increased expression of p-Smad2 and Snail; these increases were prevented by β-peltoboykinolic acid in a dose-dependent manner (Figure 6A). In addition, the expression of Snail mRNA induced by TGF-β1 was downregulated by β-peltoboykinolic acid (Figure 6B).

## 3. Discussion

In October 2014, the US Food and Drug Administration approved pirfenidone and nintedanib as drugs for the treatment of IPF. However, they received only a conditional recommendation for use, and thus, to date, no drugs are recommended for IPF [13]. Many researchers have dedicated time and effort to advancing our understanding of IPF and developing novel therapies for patients with lung fibrosis. EMT is being increasingly recognized as part of the process following epithelial stress/injury, whereby epithelial cells give rise to fibroblasts and thereby contribute to the pathogenesis of fibrosis. A large body of evidence has accumulated detailing the beneficial effects of botanic natural products against EMT. The Chinese herb *Astragalus membranaceus* reduces lung inflammation and EMT in lung cells and tissues [14]. In addition, various natural active ingredients including resveratrol, curcumin, tannic acid, and ginkgolic acid inhibit TGF-β1-induced EMT in A549 cells [15,16,17,18]. The natural active ingredient, β-peltoboykinolic acid, has been isolated from members of the Saxifragaceae family including *Aceriphyllum rossii* [19], *Chrysosplenium grayanum* Maxim. [20], and *Astilbe koreana* [21]. It has been reported to exhibit anticomplement activity [19], cytotoxicity [20,22], and protein tyrosine phosphatase 1B inhibitory effects [21]. However, the present study is the first to identify the inhibitory effect of β-peltoboykinolic acid on EMT induced by TGF-β1.

As previously mentioned, TGF-β was first described as an inducer of EMT in normal mammary epithelial cells and has since been shown to mediate EMT in vitro in several epithelial cell types including the renal proximal tubular, lens, and most recently, alveolar epithelial cells [6]. TGF-β is upregulated in the lungs of patients with IPF, and the expression of active TGF-β in the lungs of rats induces a strong fibrotic response, whereas the inability to respond to TGF-β1 affords protection from bleomycin-induced fibrosis [23]. In the present study, ARE prevented TGF-β1-induced cellular responses and these effects were attributable to ARADF and ARRDF (Figure 1 and Figure 3). In addition, treatment with a triterpene β-peltoboykinolic acid present in the CH_2_Cl_2_ fractions blocked the decrease in E-cadherin expression considerably and the increases in N-cadherin and vimentin expression induced by TGF-β1 (Figure 4). These results strongly indicate that β-peltoboykinolic acid derived from *A. rubra* promotes E-cadherin expression and suppresses N-cadherin expression during TGF-β1-induced EMT.

The pathogenesis of fibrosis is characterized by the overproduction and deposition of ECM. ECM comprises proteoglycans such as decorin and fibromodulin, fibrous proteins such as collagen, elastin, and fibrillin, adhesion molecules such as fibronectin and laminin, and several types of matrix metalloproteinases [24]. COL1 is the major fibrous protein in ECM. The uncontrolled excessive synthesis and deposition of COL1 in the lung results in progressive lung fibrosis. Moreover, COL1 promotes the downregulation of E-cadherin expression [25]. Cellular fibronectin, which is another ECM component, contributes to fibroblast activation and wound healing [26]. In the present study, β-peltoboykinolic acid effectively suppressed the production of COL1 and fibronectin induced by TGF-β (Figure 5).

Many studies have reported that the Smad pathway has a role in inducing the expression of EMT- and ECM-related genes [27,28]. The canonical Ser 465/467 phosphorylation of Smad2 by TGF-β1 increases the expression of Snail, which reduces E-cadherin expression, resulting in the destruction of cell–cell adhesion and an increase in migration ability [29]. In addition, regulation of the Smad pathway is reported to be associated with ECM production [30]. Therefore, the Smad pathway is a key regulator in fibrotic EMT. In the present study, β-peltoboykinolic acid inhibited the Ser 465/467 phosphorylation of Smad2 and the increase in Snail induced by TGF-β1 (Figure 6). A variety of natural active ingredients inhibiting EMT and ECM accumulation via intervention of the Smad pathway have been recently reported [31]. As our results are consistent with these reports, we suggest that β-peltoboykinolic acid attenuated EMT through the suppression of TGF-β1-induced Smad/Snail signaling. In addition, the suppression of ECM production by β-peltoboykinolic acid is assumed to be associated with the interruption of the canonical Smad pathway. The Ser245/250/255 phosphorylation of Smad2 as the non-canonical pathway induced the activation of MAPKs, which is associated with EMT [32,33]. Recent research has confirmed that natural active ingredients can block various pathways such as MAPK/ERK1/2, PI3K/AKT, FAK/Src, and WNT/β-catenin, can inactivate key proteins such as NF-κB, STAT3, and HIF-1α, and can also inhibit EMT at the transcriptional level [6]. In this regard, further study of the mechanisms of the inhibition of EMT by β-peltoboykinolic acid is needed to strengthen the evidence for its use as a chemotherapeutant.

In conclusion, the present study demonstrated that β-peltoboykinolic acid derived from *A. rubra* has an inhibitory effect on EMT and ECM production induced by TGF-β1. The chemical structures of various drugs are often established from the basic chemical structures of botanical active ingredients. TGF-β signaling inhibitors are generally safe and may be effective in several clinical applications, especially in cases such as end-stage cancer or IPF. In particular, several compounds targeting EMT have proceeded through clinical trials [34]. Although research on the effects of botanical active ingredients on EMT has been limited to the cell and animal levels, they are still promising for fibrosis therapy, with the consideration of the multiple effects of signals on fibrosis. Further studies on the therapeutic effects of β-peltoboykinolic acid in animal models with lung fibrosis are needed to accomplish the transition from phytochemical to drug.

## 4. Materials and Methods

### 4.1. Plant Material

Whole plants of *A. rubra* were collected in June 2017 at Pocheon, Gyeonggido, Korea. A voucher specimen was deposited at the herbarium of the School of Pharmacy, Sungkyunkwan University (SKKU-Ph-17-015).

### 4.2. Extraction and Fractionation

Whole plants of *A. rubra* (dry weight 300 g) were subjected to extraction twice with 70% EtOH at room temperature (24 °C) for 24 h, and then once at 60 °C for 5 h. The combined extracts were evaporated in vacuo to yield the 70% EtOH extract (ARE). The dried aerial part of *A. rubra* (dry weight 146 g) was subjected to extraction twice with 70% EtOH at room temperature (24 °C) for 24 h and once at 60 °C for 5 h. All extracts were combined, and the solvent was evaporated under reduced pressure to prepare the EtOH extract of the aerial part (ARAE, 12.49 g). ARAE was suspended in DW (100 mL), and the resulting solution was consecutively partitioned with dichloromethane (CH_2_Cl_2_), ethyl acetate (EtOAc), and n-butanol (n-BuOH) to yield CH_2_Cl_2_ (ARADF, 3.58 g), EtOAc (ARAEF, 0.96 g), n-BuOH (ARABF, 1.65 g), and water (ARAWF, 6.21 g) fractions. The dried rhizome of *A. rubra* (dry weight 356 g) was subjected to extraction twice with 70% EtOH at room temperature (24 °C) for 24 h, and once at 60 °C for 5 h. After extraction, the total filtrate was concentrated to dryness under reduced pressure, and the residue (ARRE, 9.49 g) was suspended in DW (70 mL). The resulting solution was successively partitioned with CH_2_Cl_2_, EtOAc, and n-BuOH to obtain CH_2_Cl_2_ (ARRDF, 1.20 g), EtOAc (ARREF, 1.65 g), n-BuOH (ARRBF, 3.07 g), and water (ARRWF, 3.53 g) fractions.

### 4.3. Isolation of β-Peltoboykinolic Acid

The CH_2_Cl_2_ fraction, which showed the strongest inhibitory effect on TGF-β1-induced EMT, was subjected to column chromatographic separation. The CH_2_Cl_2_ fraction from the rhizome (ARRDF) was fractionated on a silica gel column using stepwise elution with several mixtures of hexane and EtOAc (30:1, 10:1, 5:1, and 1:1) to yield fractions D01–D16. Fraction D11 was re-chromatographed over a RP-C_18_ column with 90% MeOH as the eluent to obtain seven fractions (D11-1–D11-7). Fraction D11-5 was further fractionated using silica gel column chromatography (hexane/EtOAc = 4:1) to give five fractions (D11-5-1–D11-5-5). β-Peltoboykinolic acid was obtained as a white solid (yield: 28.8 mg).

### 4.4. Cell Culture and Reagents

Human type 2 alveolar epithelial cells (A549) were purchased from the Korean Cell Line Bank (Seoul, Korea) and grown in RPMI 1640 (Gibco Laboratories, Grand Island, NY, USA), supplemented with 5% fetal bovine serum (Biotechnics Research Inc., Lake Forest, CA, USA) and penicillin/streptomycin (100 units/mL) at 37 °C in a humidified atmosphere with 5% CO_2_. Recombinant human TGF-β1 was purchased from R&D Systems (Minneapolis, MN, USA). Primary antibodies for the identification of N-cadherin, vimentin, E-cadherin, p-Smad2, Smad2, and Snail were purchased from Cell Signaling Technologies (Danvers, MA, USA). COL1 and fibronectin primary antibody and secondary anti-rabbit antibody were purchased from Abcam (Cambridge, MA, USA).

### 4.5. Western Blot Analysis

A549 cells were seeded into 6-well plates. After 48 h, the cells were treated with 2 ng/mL of TGF-β1 alone or with all extracts and solvent fractions. The cells were washed twice with PBS, lysed with 70 μL of RIPA buffer (#89901; Thermo Scientific, Rockford, IL, USA), and incubated on ice for 5 min. The cells were then collected into a 1.5-mL microtube by scraping, followed by incubation on ice for 30 min, with vortex mixing every 10 min. The cells were subsequently centrifuged at 14,000× *g* for 15 min at 4 °C. The supernatants were transferred to a new microtube and the protein concentration was quantified using the Micro BCA Protein Assay Kit (Pierce, Rockford, IL, USA). An equal volume of the samples was denatured by the application of a buffer containing 0.06 M Tris-HCl, 6% 2-mercaptoethanol, 2% SDS, 0.004% bromophenol blue, and 40% glycerol at 90 °C–100 °C for 6 min. Denatured total protein (15 μg) was loaded onto the gel. After the application of an electric current (80 V for 20 min and 120 V for 60 min), the proteins were electrotransferred onto a 0.2 mm PVDF membrane (Bio-Rad Laboratories, Hercules, CA, USA) using the Trans-Blot Turbo system (Bio-Rad Laboratories). Non-specific binding to the membrane was blocked by incubation in TBST containing 5% skim milk for 1 h at room temperature (24 °C) and then incubated with the appropriate antibodies. The primary antibodies (and dilution used) were rabbit anti-E-cadherin antibody (1:2000), rabbit anti-vimentin antibody (1:5000), and rabbit anti-N-cadherin antibody (1:5000); all dilutions were made in blocking buffer. Rabbit anti-Snail antibody (1:1000), rabbit anti-Smad2 antibody (1:1000), and rabbit anti-phosphorylated Smad2 (Ser 465/467) antibody (1:1000 dilution) were prepared in TBST containing 5% BSA. An anti-rabbit secondary antibody was used, diluted at 1:10,000 in blocking buffer or TBST containing 5% BSA. The membrane was incubated with ECL substrate (Bio-Rad Laboratories) for 5 min and the signals were developed onto x-ray film (JPI Healthcare, Seoul, Korea). Western blot images were quantified using ImageJ 1.52a, with the density of each band normalized to that of GAPDH.

### 4.6. COL1 Western Blot Analysis

Samples were denatured with the application of the buffer above-mentioned in the Western blot analysis at 100 °C for 13 min. Denatured total proteins (30 μg) were loaded onto the gel. After electrophoresis on a 4%–20% Mini-PROTEAN TGX precast gel (Bio-Rad Laboratories) at 120 V for 60 min, the proteins were electrotransferred onto a 0.2-mm PVDF membrane (Bio-Rad Laboratories) using the Trans-Blot Turbo system (Bio-Rad Laboratories). Non-specific binding to the membrane was blocked by incubation of the membrane in TBST containing 5% skim milk for 1 h at room temperature (24 °C); subsequently, the membrane was incubated with the appropriate antibodies. The primary antibody used was a rabbit anti-COL1 antibody at a 1:1000 dilution in blocking buffer, and the secondary antibody used was an anti-rabbit antibody at a 1:1000 dilution in blocking buffer. Western blot images were quantified using ImageJ, with the density of each band normalized to that of GAPDH.

### 4.7. Migration Assay

Scratches were introduced using a wound maker, which created wounds of equal width. A549 cells (3 × 10^4^ cells/well) were plated onto a 96-well ImageLock microplate (Essen BioScience, Ann Arbor, MI, USA) and treated with TGF-β1 (5 ng/mL), either alone or with samples. The cells were washed twice with media to remove debris. Wound images were automatically acquired and recorded by the IncuCyte ZOOM imaging system (Essen Biosciences) using time-lapse bright-field microscopy.

### 4.8. Quantitative Real-Time PCR (qRT-PCR)

Total cellular RNA was isolated from the A549 cells exposed to β-peltoboykinolic acid with TGF-β1 using the RNAiso reagent (Takara Bio Inc., Shiga, Japan) in accordance with the manufacturer’s instructions. Total RNA was reversely transcribed into cDNA with the AccuPower^®^ RocketScript RT-PCR premix (Bioneer, Daejeon, Korea). The reaction was performed in accordance with the manufacturer’s instructions. Gene expression was determined using an SYBR premix Ex-Taq™ II (Takara Bio Inc.). The PCR amplification was performed using a CFX Connect Real-Time System (Bio-Rad Laboratories). The relative amount of each cDNA was determined by the 2^−ΔΔCt^ method, with GAPDH as the internal control. Primers for qRT-PCR are shown in Appendix A.

### 4.9. Statistical Analysis

Each in vitro assay was performed at least three times. The data were analyzed using Excel (Microsoft, Redmond, WA, USA) and expressed as the mean ± standard deviation. Statistical analysis was computed using SPSS version 21.0 (SPSS, Chicago, IL, USA). Differences between groups were assessed by the Duncan’s post hoc test following one-way analysis of variance. Statistical significance was accepted at *p* < 0.01 or 0.05.

## Figures and Tables

**Figure 1 molecules-24-02573-f001:**
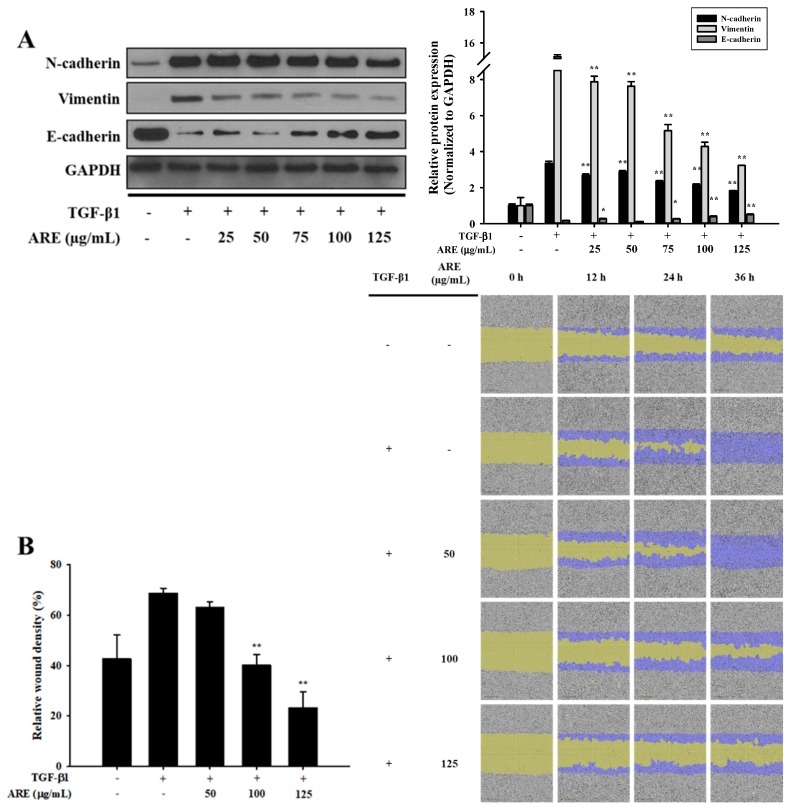
*Astilbe rubra* extract inhibited TGF-β1-induced epithelial-to-mesenchymal (EMT) in A549 cells. A549 cells were co-treated with TGF-β1 and *A. rubra* extract (ARE) for 48 h. (**A**) The expression of N-cadherin, vimentin, and E-cadherin proteins were analyzed using Western blot analysis, with GAPDH expression used as a loading control. ImageJ was used for the quantification of the Western blots. (**B**) A549 cells were plated into 96-well plates and treated with TGF-β1 (5 ng/mL) or co-treated with ARE. The cell mobility was analyzed by IncuCyte ZOOM over 36 h at 10× magnification through a phase-contrast objective lens. The bar graph represents the wound density at 24 h exposure. All data are shown as mean ± S.D., *n* = 3. ** *p* < 0.01 vs. the TGF-β1-treated group.

**Figure 2 molecules-24-02573-f002:**
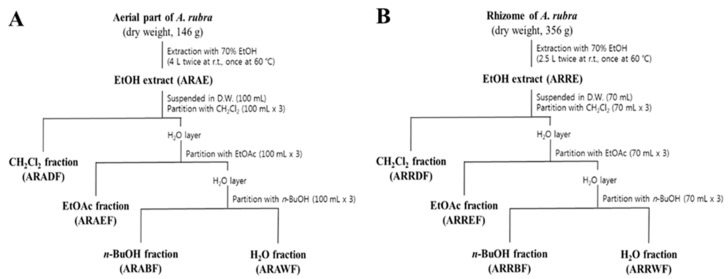
Extraction and fractionation scheme for *Astilbe rubra*. (**A**) Extraction and fractionation of the aerial part. (**B**) Extraction and fractionation of the rhizome. The dried samples were subjected to extraction separately with 70% ethanol. The extracts of the aerial parts and rhizome (ARAE and ARRE, respectively) were consecutively partitioned with dichloromethane (CH_2_Cl_2_), ethyl acetate (EtOAc), and butanol (n-BuOH) to give four solvent fractions.

**Figure 3 molecules-24-02573-f003:**
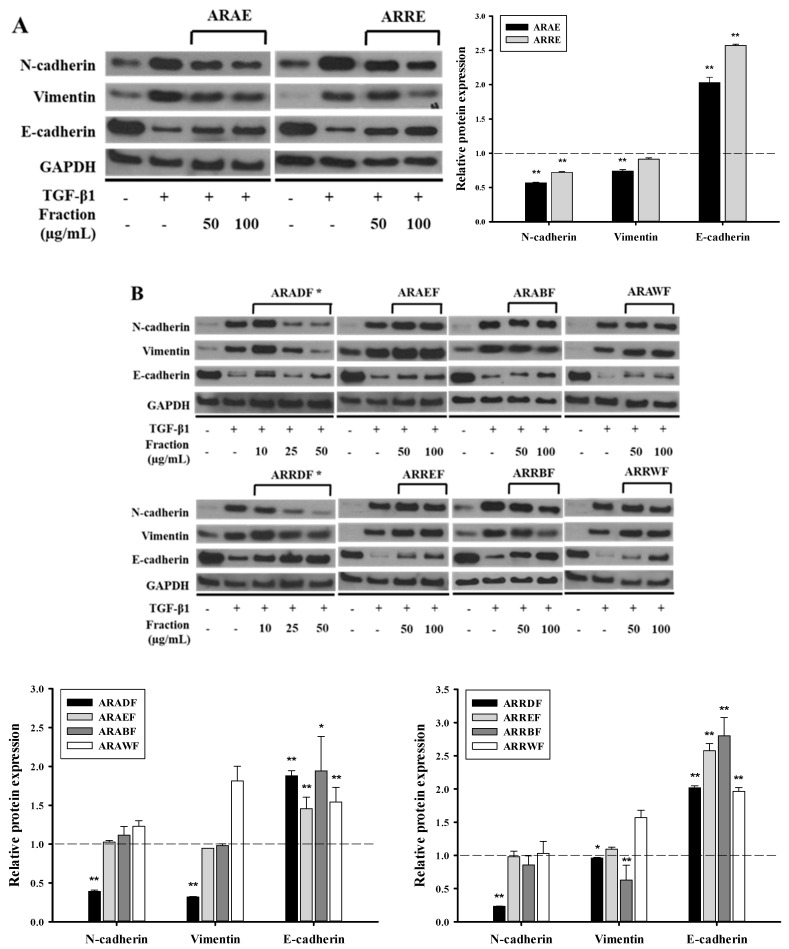
Inhibitory effect of fractions derived from *Astilbe rubra* on TGF-β1-induced epithelial-to-mesenchymal (EMT). A549 cells were co-treated with TGF-β1 (2 ng/mL) along with the ethanol extracts (**A**) and the fractionated samples (**B**) of *A. rubra* for 48 h. All samples, except for the CH_2_Cl_2_ fractions, were treated at concentrations of 50 and 100 μg/mL. The CH_2_Cl_2_ fractions were treated at concentrations of 10, 25, and 50 μg/mL. Protein expression was investigated using Western blot analysis, with GAPDH used as a loading control. * shows the strongest inhibitory effect on the TGF-β1-induced EMT of all the fractions tested. The bar graphs represent the quantification of the Western blots using ImageJ. * *p* < 0.05, ** *p* < 0.01 vs. the TGF-β1-treated group (dot-line). ARAE, EtOH extract of the aerial part; ARADF, CH_2_Cl_2_ extract of the ARAE; ARAEF, EtOAc extract of the ARAE; ARABF, n-BuOH extract of the ARAE; ARAWF, water extract of the ARAE; ARRE, EtOH extract of the rhizome; ARRDF, CH_2_Cl_2_ extract of the ARRE; ARREF, EtOAc extract of the ARRE; ARRBF, n-BuOH extract of the ARRE; and ARRWF, water extract of the ARRE.

**Figure 4 molecules-24-02573-f004:**
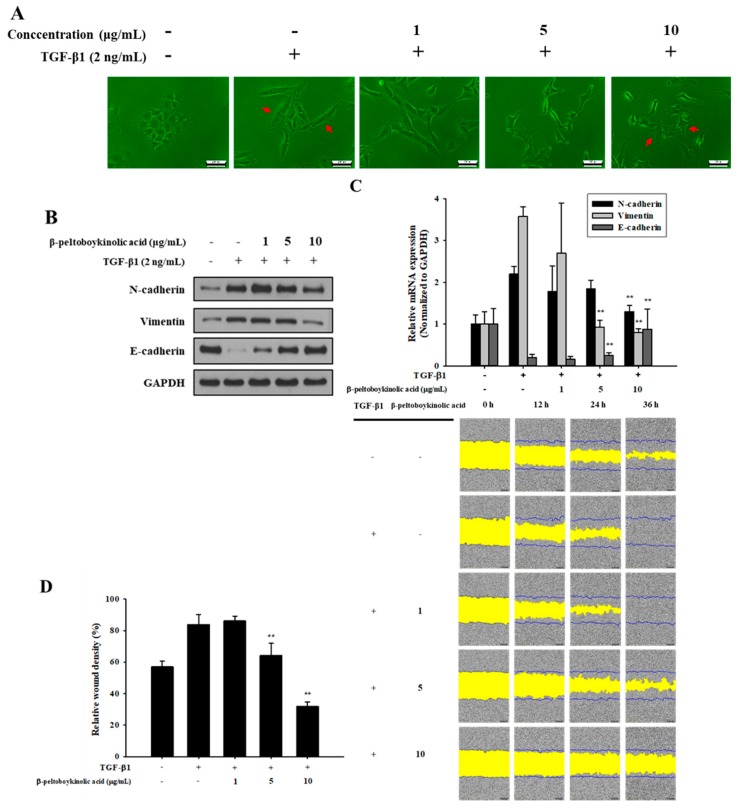
Inhibitory effect of β-peltoboykinolic acid on TGF-β1-induced epithelial-to-mesenchymal (EMT) in A549 cells. A549 cells were treated with TGF-β1 or co-treated with β-peltoboykinolic acid for 48 h. (**A**) The morphology of A549 cells was observed using a phase-contrast microscope. Arrows indicate the difference between cell morphologies. Scale bar = 200 μm. (**B**) The expression of N-cadherin, vimentin, and E-cadherin proteins were analyzed using Western blot analysis, with GAPDH expression used as a loading control. (**C**) A549 cells were treated with TGF-β1 (2 ng/mL) or co-treated with β-peltoboykinolic acid (1, 5, and 10 μg/mL) for 24 h. The expression of N-cadherin, vimentin, and E-cadherin mRNA were analyzed using qRT-PCR. Fold change was calculated using 2^−ΔΔCt^ relative quantitative analysis. (**D**) A549 cells were treated with TGF-β1 (5 ng/mL) or co-treated with β-peltoboykinolic acid (1, 5, and 10 μg/mL). The cell mobility was analyzed by IncuCyte ZOOM over 36 h at 10× magnification through a phase-contrast objective lens. The bar graph represents the wound density at 24 h exposure. All data are shown as mean ± S.D., *n* = 3. ** *p* < 0.01 vs. the TGF-β1-treated group.

**Figure 5 molecules-24-02573-f005:**
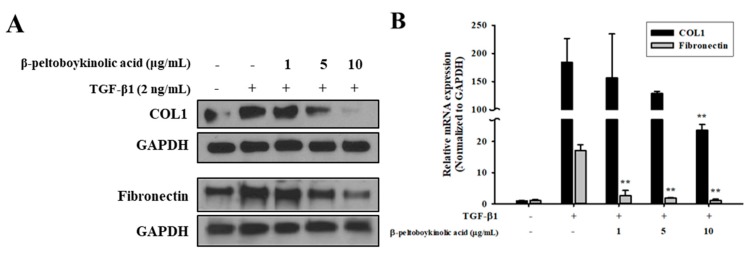
Inhibitory effect of β-peltoboykinolic acid on extracellular matrix (ECM) production induced by TGF-β1. A549 cells were treated with TGF-β1 (2 ng/mL) or co-treated with β-peltoboykinolic acid (1, 5, and 10 μg/mL) for 48 h. (**A**) The protein expression of type I collagen (COL1) and fibronectin were analyzed by Western blotting, with GAPDH used as a loading control. (**B**) A549 cells were treated with TGF-β1 (2 ng/mL) or co-treated with β-peltoboykinolic acid for 24 h. The mRNA expression of COL1 and fibronectin were analyzed using qRT-PCR. Fold change was calculated using 2^−ΔΔCt^ relative quantitative analysis. All data are shown as mean ± S.D., *n* = 3. ** *p* < 0.01 vs. the TGF-β1-treated group.

**Figure 6 molecules-24-02573-f006:**
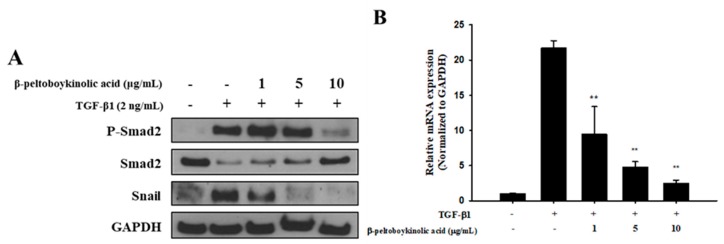
Effect of β-peltoboykinolic acid on the Smad pathway in A549 cells treated with TGF-β1. (**A**) A549 cells were treated with TGF-β1 (2 ng/mL) or co-treated with β-peltoboykinolic acid (1, 5, and 10 μg/mL) for 48 h. Protein expression was analyzed using Western blotting, with GAPDH used as a loading control. (**B**) A549 cells were treated with TGF-β1 (2 ng/mL) or co-treated with β-peltoboykinolic acid (1, 5, and 10 μg/mL) for 24 h. The expression of Snail mRNA was analyzed using qRT-PCR. Fold change was calculated using 2^−ΔΔCt^ relative quantitative analysis. All data are shown as mean ± S.D., *n* = 3. ** *p* < 0.01 vs. the TGF-β1-treated group.

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
