# Peer review of "β-Peltoboykinolic Acid from Astilbe rubra Attenuates TGF-β1-Induced Epithelial-to-Mesenchymal Transitions in Lung Alveolar Epithelial Cells"

_molecules, 2019, doi:10.3390/molecules24142573_

Round 1
Reviewer 1 Report
This is a well written manuscript, that describes the activity of Astilbe rubra on the TGF-β1-induced epithelial-to-mesenchymal transitions in lung alveolar epithelial cells.
Results are sound and appropriately presented.
There are some minor considerations to do.
It is not clear the treatment with fractioned extracts: is the solvent eliminated and dry fractions suspended in water as in the whole ethanol extracts? If not, control treatments using solvent alone should be done and showed.
Figure 1B: please show the images, in fig 1 or in the supplementary file.
Figure 4A: there is an error on TGF-b1 marks; as indicated now, there are not TGF-b1 treatments. The images are not clear; an image editing to improve contrast and definition should be considered; please add arrows to highlight the morphological details typical for EMT.
Supplementary material
Figure S1 B: it is shown the same graph twice
Reviewer 2 Report
In this manuscript, Band and co-authors present original data identifying a role for Astilbe Rubra derived B-peltoboykinolic acid in TGF-beta mediated EMT and ECM production in lung epithelial cells. While showing potentially interesting data the mechanistic insights into the signalling pathway are limited.
Major comments:
1. There are several major grammatical errors throughout the text and most cases sentences are not correct and this will diminish the accessibility of the manuscript. The authors should find someone who can carefully edit the English throughout the entire manuscript, in order to improve the clarity and precision of the writing.
2. What concentration of CH2Cl2 fractions were used in figure 3? It is important to know whether a higher concentration was used from them fraction - As they were defined as the most sensitive to inhibiting TGF-beta mediated EMT and lead to subsequent fractionation experiments.
3. Figure 3 – an * is used to identify which extracts have the strongest inhibition however this is not so clear between some of the blots – in the methods it has been indicated that densitometry has been completed I would suggest discussing these numbers as fold change in the results section of 2.2 to further strengthen the data and/or add normalised densitometry as supplementary data
a. This would also be of benefit for Figure 1A as a dose dependent inhibition cannot be visualised from the WB
4. The authors will need to elaborate on which residues there Smad antibodies target as different residues of the Smads are associated with different signalling pathways. Non-canonical Smad (Smad linker region phosphorylation) is associated with EMT (Ooshima et al 2019).
5. Can you please expand on the potential mechanisms which B-PA is inhibited TGF-beta mediated pSmad2, EMT and ECM
Minor comments
1. Figure 1 A used 2ng/ml of TGFB and Figure B used 5ng/ml please expand on why different concentrations were used
2. Figure 5A the western blot used for COL1 is not suitable and should be replaced – the bands cannot be seen in lanes 4 and 5 and these make a very important part of the conclusion
Round 2
Reviewer 1 Report
The manuscript has been modified as suggested, and it is now acceptable for publication.
Reviewer 2 Report
The authors have satisfactorily responded to my comments